# Assessment of microfilaremia in 'hotspots' of four lymphatic filariasis endemic districts of Nepal during post-MDA surveillance

Pramod Kumar Mehta *, Mahendra Maharjan *

Central Department of Zoology, Institute of Science and Technology, Tribhuvan University, Kirtipur, Nepal

* pramodmehta89@yahoo.com (PKM); mahendra.maharjan@cdztu.edu.np, maharjan.m@gmail.com (MM)

**Data Availability Statement:** The authors confirm that all data underlying the findings are fully available without restriction. All relevant data are

## Abstract

### Background

The lymphatic filariasis (LF) elimination program in all sixty-three endemic districts of Nepal was based on annual mass drug administration (MDA) using a combination of diethylcarbamazine (DEC) and albendazole for at least 5 years. The MDA program was started in the Parsa district of the Terai region and at least six rounds of MDA were completed between 2003 and 2017 in all filariasis endemic districts of Central Nepal. Transmission Assessment Survey (TAS) report indicated that circulating filarial antigen (CFA) prevalence was below the critical value i.e., $\leq 2\%$ in selected LF endemic districts of Central Nepal. Based on the TAS report, antigen-positive cases were found clustered in the foci of those districts which were considered as "hotspots". Hence the present study was designed to assess microfilaremia in hotspots of four endemic districts of Central Nepal after the MDA program.

### Methodology and principal findings

The present study assessed microfilaremia in hotspots of four endemic districts i.e. Lalitpur and Dhading from the hilly region and Bara and Mahottari from the Terai region of Central Nepal. Night blood samples (n = 1722) were collected by finger prick method from the eligible sample population irrespective of age and sex. Community people's participation in the MDA program was ensured using a structured questionnaire and chronic clinical manifestation of LF was assessed using standard case definition. Two districts one each from the hilly region (Lalitpur district) and Terai region (Bara district) showed improved microfilaria (MF) prevalence i.e. below the critical level (<1%) while the other two districts are still over the critical level. There was a significantly high prevalence of MF in male (p = <0.05) and $\geq 41$ years of age group (p = <0.05) community people in the hotspots of four endemic districts. People who participated in the previous rounds of the MDA program have significantly low MF prevalence. The upper confidence limit of MF prevalence in all hotspots of four districts was above the critical level (>1%). Chronic clinical manifestation of LF showed significant association with the older age group ($\geq 41$ years) but not with sex.

within the paper and its Supporting Information files.

**Funding:** PKM has received a PhD research grant from the University Grants Commission of Nepal (Award no. PhD/74-75 S and T-17). None of the authors have received a salary from the funders. The funders had no role in study design, data collection and analysis, decision to publish, or preparation of the manuscript.

**Competing interests:** The authors have declared that no competing interests exist.

## Conclusions

The study revealed LF transmission improved in hotspots of two districts while continued in others but the risk of LF resurgence cannot be ignored since the upper confidence level of MF prevalence is over 1% in all the hotspots studied districts. High MF prevalence is well correlated with the number of MDA rounds but not with the MDA coverage. Community people involved in MDA drug uptake in any previous and last rounds have significantly less MF infection. Hence it is recommended that before deciding to stop the MDA rounds it is essential to conduct the MF survey at the hotspots of the sentinel sites.

### Author summary

Lymphatic filariasis is a neglected tropical disease that causes disability to human beings and is caused by a group of filarial nematodes such as *W. bancrofti*, *Brugia malayi*, and *Brugia timori* while these parasites are transmitted by different species of vector mosquitoes. Nepal government started the MDA program in 2003 and has completed 6–11 rounds of anti-filarial medicine with DEC and albendazole between 2007 and 2022 in selected endemic districts of Central Nepal. TAS report showed a low level of infection (<2%) in all study districts. Interestingly we found that the antigen-positive cases were clustered in foci. We defined those circulating filarial antigen (CFA) positive clustered areas as hotspots and assessed MF infection among the adult community people after the MDA program. The result revealed evidence of MF prevalence above threshold in the hotspots of two districts one each from the hilly region (Dhading district) and Terai region (Mahottari district) even after the completion of the MDA program. In those areas, alternative treatment strategies should be employed to reduce the active infection which can serve as reservoirs for recrudescence transmission. Hence it is urgently needed to identify and screen MF infection among community individuals of hotspots after completion of the MDA program even in other districts.

## Introduction

Lymphatic Filariasis (LF) is a neglected tropical disease caused by a group of filarial nematodes most commonly by *Wuchereria bancrofti* and is transmitted by *Culex*, *Anopheles*, and *Aedes* species of mosquitoes [1,2]. LF is commonly known as elephantiasis which causes disability and disfiguring to human beings due to blocking the lymph vessels by dead microfilariae [3,4].

LF is a major public health problem in tropical and sub-tropical countries of the world. It was estimated that approximately 120 million people from more than 83 tropical and sub-tropical countries were affected by LF along with one billion people at risk of infection in 2000 [5]. Out of those at risk of infection, 65% were from Southeast Asia, 30% were from Africa, and the remaining lived in other tropical countries of the world [6,7]. But recently WHO estimated over 882 million people remained threatened in 44 countries worldwide [8].

The Global Program to Eliminate Lymphatic Filariasis (GPELF) was launched in 2000 and achieved significant progress in 60 countries by 2015 [9]. The program was based on the application of annual MDA with a combination of albendazole plus either diethylcarbamazine (DEC) or Ivermectin to all eligible populations in endemic areas with effective treatment

coverage of >65% [4,7,10]. In Nepal, the combination of DEC and albendazole was recommended for at least five years to eliminate LF in the endemic districts of the country.

The expected number of MDA rounds during the elimination program depends on the baseline prevalence of infection, MDA compliance and regimens during TAS [10,11]. It has been shown that the reasons for MF persistence in the community are due to non-compliance with medication and poor coverage of MDA [12–16].

Nepal is one of the filariasis endemic country with recorded infection in 63 districts while the remaining 14 districts are unlikely to be endemic because of their mountainous geographical location [17,18], Nepal government had formulated National Task Force (2003-2020AD) and the MDA program was started in 2003 to interrupt the transmission of microfilariae in the community by 2020. Six rounds of MDA were completed in all endemic districts in 2017. TAS is a surveillance tool to determine the LF infection levels are sustained below the critical cut-off value. TAS should be carried out in the age group of 6–7 years children after 5–6 rounds of MDA with drug coverage of 65% or above. If TAS results showed antigen prevalence ≤2% or MF prevalence ≤1% in community individuals indicated that the MDA round could be stopped. The TAS report showed antigenic prevalence was < 2% in 31 districts in 2018 [19]. Nepal government couldn't achieve the LF elimination goal by 2020 in some districts based on a series of TAS hence the MDA program was extended with the new LF elimination target by 2030. In Central Nepal, MDA was stopped in Dhading and Mahottari districts after six rounds while in Lalitpur and Bara districts, it was stopped after eight and eleven rounds respectively. We have analyzed the status of LF infection in these districts based on baseline prevalence and a series of TAS reports. The baseline prevalence of Dhading and Mahottari districts had a high prevalence of LF compared to Bara and Lalitpur districts. TAS report indicated CFA prevalence ranged from 0.2 to 1.2% (Table 1).

Although CFA prevalence during TAS indicated below the critical level, we found that the cases were clustered in the community. Those CFA clusters were considered hotspots and hypothesized that there could be the existence of MF carriers. Thus this study was proposed to assess microfilaremia prevalence in hotspots of four endemic districts of Central Nepal.

## Methods

### Ethics statements

Ethical approval was retrieved from the Nepal Health Research Council (NHRC/Reg.no. 629/2018). Permission for conducting the study was received from the local government. Written consent forms and information sheets were provided to participants (or parents/guardians) in the local language and written consent was obtained from the participants (or parents/guardians).

**Table 1. Reported LF prevalence during various stages of assessment in four districts of Central Nepal.**

| Region | Districts | Population | LF antigen prevalence (%) | | | | | | | |
|--------|-----------|-----------|-----------|------------|-----------|------------|-----------|------------|-----------|------------|
| | | | Baseline prevalence | | Pre-TAS | | TAS I | | TAS II | |
| | | | Year | Prevalence | Year | Prevalence | Year | Prevalence | Year | Prevalence |
| Hilly | Lalitpur | 548,401 | 2008 | 1.06 | 2016 | 0.66 | 2017 | 0.21 | 2019 | 0.18 |
| | Dhading | 322,751 | 2001 | **14.7*** | 2012 | 0.8 | 2013 | 0.8 | 2017 | 0.41 |
| Terai | Bara | 743,950 | 2001 | 0.6 | 2012 | 0.0 | 2013 | 1.2 | 2017 | 1.2 |
| | Mahottari | 705,838 | 2001 | **2.43*** | 2012 | 0.0 | 2013 | 0.6 | 2017 | 0.7 |

*, Symbols indicate antigen prevalence above the critical cut-off value (≤ 2%).

## Study area

TAS report of Epidemiology and Disease Control Division (EDCD) 2017–2018 indicated comparatively high CFA prevalence in four districts of Central Nepal, two districts (Lalitpur (0.21%) and Dhading (0.41%) from the hilly region and two districts, Bara (1.2%) and Mahottari (0.7%) from Terai region. Due to CFA prevalence below the critical level MDA has already been stopped. On the basis of TAS1, we analyzed the CFA prevalence data in each of the sentinel sites and identified the distribution of the cases in the distinct clusters. We found those clusters at Bungmati and Dukuchhap area (local name) of Lalitpur district (five cases), Salyantar of Dhading district (nine cases), Khairawa and Ammadar of Bara district (six cases) and Matihani of Mahottari district (eight cases), which were considered hotspots and MF survey was conducted. Furthermore, we assessed the MDA coverage and total MDA rounds completed between 2007 and 2022 in the selected endemic districts of Central Nepal (Fig 1).

## Study population and sampling

We used Geographic Information System (GIS) data from the Department of Commerce of Nepal for the identification of households in the hotspots of four districts of Central Nepal.

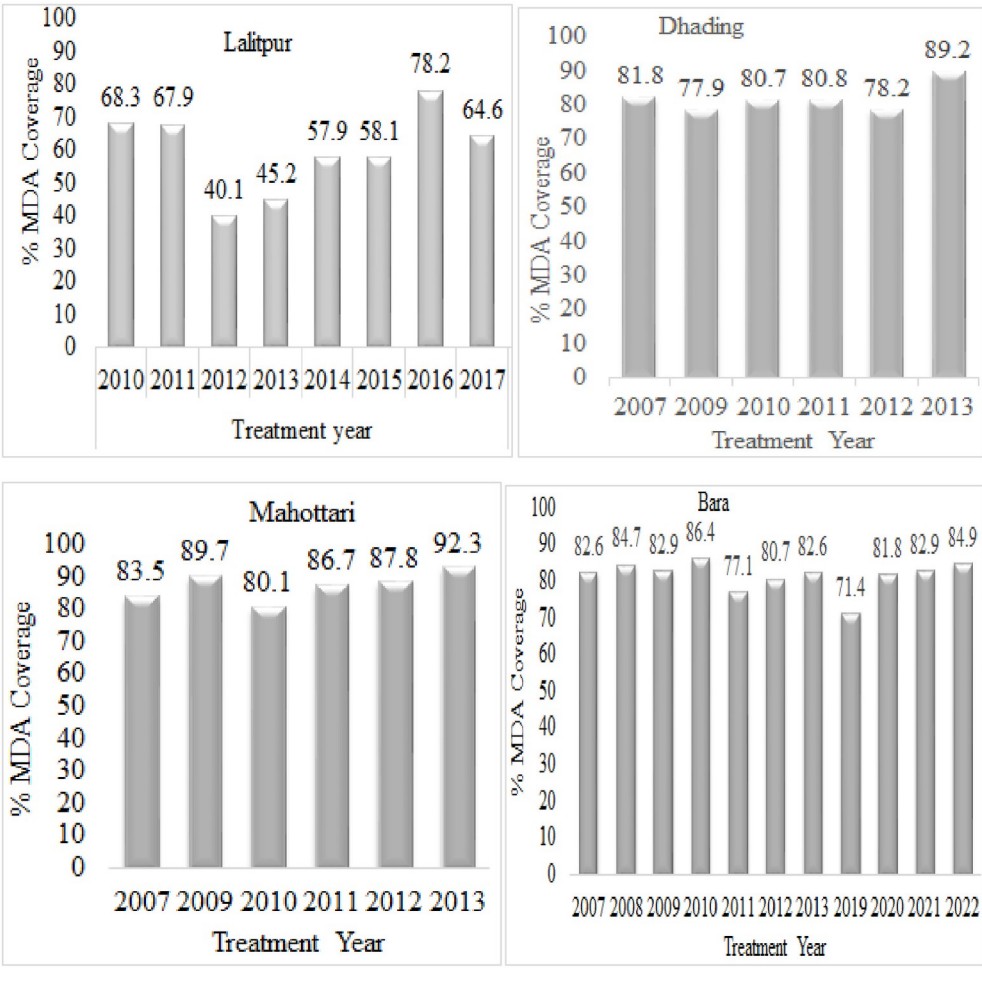

**Fig 1. Reported treatment coverage during diethylcarbamazine and albendazole MDA intervention in hotspots of the study area.**

Our target population was the age group of 10 years and above since the below 10 years age group have never participated in the MDA program and were born after the MDA program. The required sample size from hotspots of each district was determined using the formula:

n = $Z^2pq/L^2$

 Where, n = number of samples

 Z = std. normal deviate (1.96 in 95% confidence interval)

 p = prevalence of lymphatic filariasis (Probability of LF prevalence is 50%)

 q = 1-p = 1–0.5 = 0.50

 l = Error of margin = 5% = 0.05

 So, n = $(1.96)^{2*}0.50*0.50/(0.05)^2$ = 385

 Non response rate = 10% = 385*10/100 = 38.5

 So the actual number of samples for microfilaria detection was (n) = 385+38.5 = 423.5~424.

 So 424 samples were targeted for the MF survey from hotspots of each selected districts of Central Nepal.

Assuming at least three target group populations in each household, we selected a total of 724 households from the study area. From each of the hotspots of Lalitpur and Dhading districts from the hilly region, 186 and 174 households were selected respectively. Similarly, 141 and 223 households were selected from Bara and Mahottari districts from the Terai region to obtain the required sample size from each district. ArcGIS software was used to locate selected households and generate printed maps for the MF survey.

The members from the selected houses were enlisted by making a door-to-door visit and estimated the required number of population from each district. A total of 2285 eligible individuals from 724 households were enumerated. People who were outside the house during the study period and who didn't give consent to participate in the study were excluded. Finally, a total of 1722 individuals were screened for the MF.

Using the structured questionnaire MDA compliance was assessed along with their demographic information. Each of them was asked whether they had swallowed anti-filarial drugs in the last round of MDA and drug uptake in any previous rounds of MDA. Chronic clinical manifestations of LF among the sampled population such as elephantiasis and hydrocele were privately assessed by authorized medical officers of the respective health post. Based on standard case definition, long-term swelling of legs, hands, breasts etc. was considered elephantiasis and scrotum swelling was considered hydrocele (S1 Fig).

## Night blood survey for MF prevalence

A total of 1722 night blood samples were collected between 10 PM to 4 AM due to the nocturnal periodicity of *Wuchereria bancrofti*. Written consent was taken from all the individuals and parents of children below 15 years. The standard finger-prick method was applied briefly, the third or fourth finger of the left hand was cleaned with the alcohol putting the palm upward, air dried and pricked with a sterile lancet towards the internal side of the finger. The first drop of the blood was wiped away with dry cotton wool by applying gentle pressure to the finger. Three drops of blood were collected onto three portions of the single slide (S2 Fig).

Each drop of the blood was spread with the edge of another clean slide making a thick blood film. The slides with blood film were air-dried for about 12 to 24 hours and transported to the laboratory for further analysis. Slide numbers were marked in the corner of the slide using a lead pencil. In the laboratory, blood films were dehaemoglobinized using distilled water for about three minutes. Blood films were air dried, and fixed in acid-alcohol (2 parts of the con. HCL + 98 parts of methyl alcohol). Stained with Giemsa stain for 1–2 minutes and washed in distilled water.

After air drying the blood smear was examined under a compound microscope using a 10X objective. The entire blood smear was examined systematically from one end to the other end. The species of MF was confirmed using 40X objectives (S3 Fig).

The results were entered along with the number of MF against the ID number in the log book. Technical staff cross-examined all the positive and 5% of the negative slides. MF-positive cases were treated with a standard dose of DEC (6 mg/kg body weight) for 12 days under the supervision of an authorized medical officer. (Fig 2).

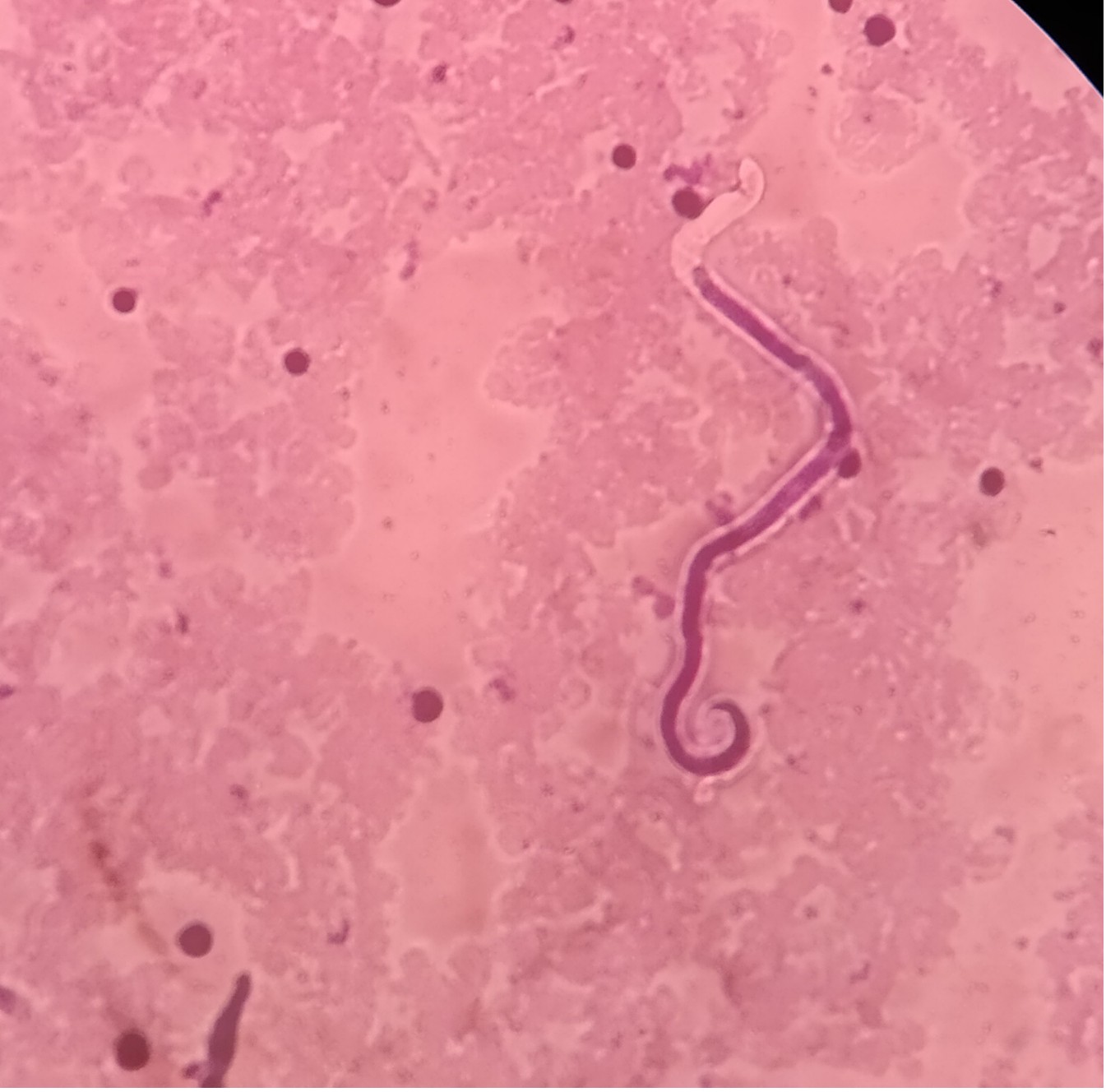

**Fig 2. Blood film showing microfilaria of *W. bancrofti*.**

## Statistical analysis

Data were entered in Excel spreadsheets (Microsoft Excel 2007) and subsequently analyzed with Minitab 17 version 19.2.0. Blood test results for MF, the presence of hydrocele or elephantiasis, demographic characteristics, and MDA compliance (site-based, district-wise) were compared by using the Chi-square test and Fisher's exact test while a p-value $\leq 0.05$ was considered statistically significant. The lower and upper limits of the 95% CI for the prevalence of MF were calculated.

## Results

### Demographic characteristics of the eligible and sampled population

A total of 2285 individuals were found to be eligible for the MF survey in hotspots of four endemic districts of Central Nepal. Among them, 1722 i.e. 75% of individuals were available and screened for MF using night blood collection by finger prick method. The maximum eligible population was covered from the Lalitpur and Bara districts. The average mean age of the individuals was 38 years in between the age group of 10 to 87 years. A maximum of 54% of females were screened for MF screening (Table 2).

### Microfilariae and disease burden in hotspots of Central Nepal

**Prevalence of microfilariae and chronic clinical manifestation of LF in Central Nepal.** The overall prevalence of microfilariae in hotspots of four endemic districts was 2.9%. The result revealed that MF infection in males was significantly higher compared to females but no significant difference was observed between the age group $\leq 40$ and $\geq 41$ years. The MF prevalence in two districts one each from the hilly region (Lalitpur district) and the Terai region (Bara district) is lower than a 1% cut off WHO recommends to stop MDA. But in the

**Table 2. Age and sex-wise eligible and sampled population in hotspots of four districts of Central Nepal.**

| Demographic characteristics | | | Region | | | | Total no. (%) |
|---|---|---|---|---|---|---|---|
| | | | Hilly | | Terai | | |
| | | | Lalitpur[#] No. (%) | Dhading[β], No. (%) | Bara[γ,] No. (%) | Mahottari[α], No. (%) | |
| Eligible population | Gender | Male | 279 (54.4) | 244 (42.8) | 210 (44.8) | 333 (45.4) | 1066 (46.7) |
| | | Female | 234 (45.6) | 326 (57.2) | 259 (55.2) | 400 (54.6) | 1219 (53.3) |
| | | Total | 513 | 570 | 469 | 733 | 2285 |
| | Age group (Years) | $\leq 40$ | 286 (55.8) | 298 (52.3) | 262 (55.9) | 396 (54.1) | 1242 (54.4) |
| | | $\geq 41$ | 227 (44.2) | 272 (47.7) | 207 (44.1) | 337 (45.9) | 1043 (45.6) |
| | | Total | 513 | 570 | 469 | 733 | 2285 |
| Sampled population | Gender | Male | 175 (44.5) | 175 (40.6) | 185 (47.1) | 226 (44.8) | 761 (44.2) |
| | | Female | 219 (55.5) | 256 (59.4) | 208 (52.9) | 278 (55.2) | 961 (55.8) |
| | | Total | 394 | 431 | 393 | 504 | 1722 |
| | Age group (Years) | $\leq 40$ | 223 (56.6) | 235 (54.5) | 241 (61.3) | 330 (65.5) | 1029 (59.8) |
| | | $\geq 41$ | 171 (43.4) | 196 (45.5) | 152 (38.7) | 174 (34.5) | 693 (40.2) |
| | | Total | 394 (76.8) | 431 (75.6) | 393 (83.8) | 504 (68.8) | 1722 (75.4) |

Note

[#] Dukuchhap and Bungmati of Lalitpur district

[β] Salyantar of Dhading district

[γ] Ammadar and Khairawa of Bara district

[α] Matihani of Mahottari district.

**Table 3. Prevalence of microfilaremia and chronic clinical manifestation of lymphatic filariasis in Central Nepal.**

| Characteristic | | No. (%) total | No.(%) with MF[95% CI] | X² Value (p-value) | No. (%) with hydrocele* | Z- value (p-value) | No. (%) with elephantiasis | X² Value (p-value) |
|---|---|---|---|---|---|---|---|---|
| Gender | Male | 761(44.2) | 29 (3.8) (2.6–5.4) | 3.749 (0.053) | 68 (8.9) | - | 30 (3.9) | 0.049 (0.825) |
| | Female | 961(55.8) | 21(2.2) (1.4–3.3) | | - | | 40 (4.2) | |
| | Total | 1722 | 50 (2.9) (2.2–3.8) | - | - | - | 70 (4.1) | - |
| Age groups in years | ≤40 | 1029 (59.8) | 25 (2.4) (1.6–3.6) | 1.919 (0.166) | 32(7.0) | 2.17 **(0.027)** | 12 (1.2) | 50.173 **(0.000)** |
| | ≥41 | 693 (40.2) | 25 (3.6) (2.4–5.3) | | 36(11.8) | | 58 (8.4) | |
| | Total | 1722 | 50 (2.9) (2.2–3.8) | - | 68 (8.9) | - | 70 (4.1) | - |
| Hilly districts | Lalitpur# | 394 (47.8) | 1 (0.3) (0.01–2.5) | 19.541 **(0.001)** | 24 (13.7) | 1.17 (0.307) | 28 (7.1) | 0.006 (0.939) |
| | Dhading^β | 431 (52.2) | 25 (5.8) (3.6–8.1) | | 32 (18.3) | | 30 (7.0) | |
| | Total | 825 | 26 (3.2) (2.1–4.6) | - | 56 (16.0) | - | 58 (7.1) | - |
| Terai districts | Bara^γ | 393 (43.8) | 2 (0.5) (0.06–1.8) | 12.012 **(0.001)** | 8 (4.3) | 1.48 (0.148) | 7 (1.8) | 1.013 (0.314) |
| | Mahottari α | 504 (56.2) | 22 (4.4) (2.8–6.5) | | 4 (1.8) | | 5 (1.0) | |
| | Total | 897 | 24 (2.7) (1.7–4.0) | - | 12 (2.9) | - | 12 (1.3) | - |

*Denominator included only males

# Dukuchhap and Bungmati of Lalitpur district

β Salyantar of Dhading district

γ Ammadar and Khairawa of Bara district

α Matihani of Mahottari district.

remaining two districts, Dhading district belonging to the hilly region and Mahottari district belonging to the Terai region showed unexpectedly high MF prevalence i.e. 5.8% and 4.4% respectively. Even though MF prevalence is below the critical level in two districts, the upper confidence limit of MF prevalence in all hotspots of four endemic districts had above critical level (≥ 1%) (Table 3).

Chronic clinical manifestation of LF was highly prevalent in hotspots of all four endemic districts. Hydrocele and elephantiasis both cases were found significantly high in the hilly region compared to the Terai region but the disease burden was not significantly different in between the districts of both the hilly and Terai regions. Similarly, both chronic disease burden was found to be significantly high in the age group ≥ 41 i.e. older age group (Table 3).

**District-wise prevalence of MF in different age and sex groups.** In three districts, Dhading, Bara and Mahottari maximum individuals were screened for MF among the age group 21–30 years. MF infection was observed in all age groups individuals in Dhading districts while the infection was not observed in the age group above 60 years in Mahottari. In both districts high MF prevalence was found in between the age group 41–60 years. Only one and two individuals were found infected with MF in the Lalitpur and Bara districts respectively. Comparatively less infection of MF was found in the age group below 41 years (Fig 3).

Maximum female individuals were covered during MF screening in all four districts. Sex-wise prevalence of MF in males of Dhading district has a high prevalence i.e. 10.3%. While in the Mahottari district belonging to the Terai region, both male and female individuals were almost equally infected (Fig 4).

## MDA compliance concerning MF infection and chronic clinical manifestation

All 1722 individuals screened for MF were enrolled in a questionnaire survey with their participation in the MDA program. In all hotspots of four districts, MDA coverage among the study

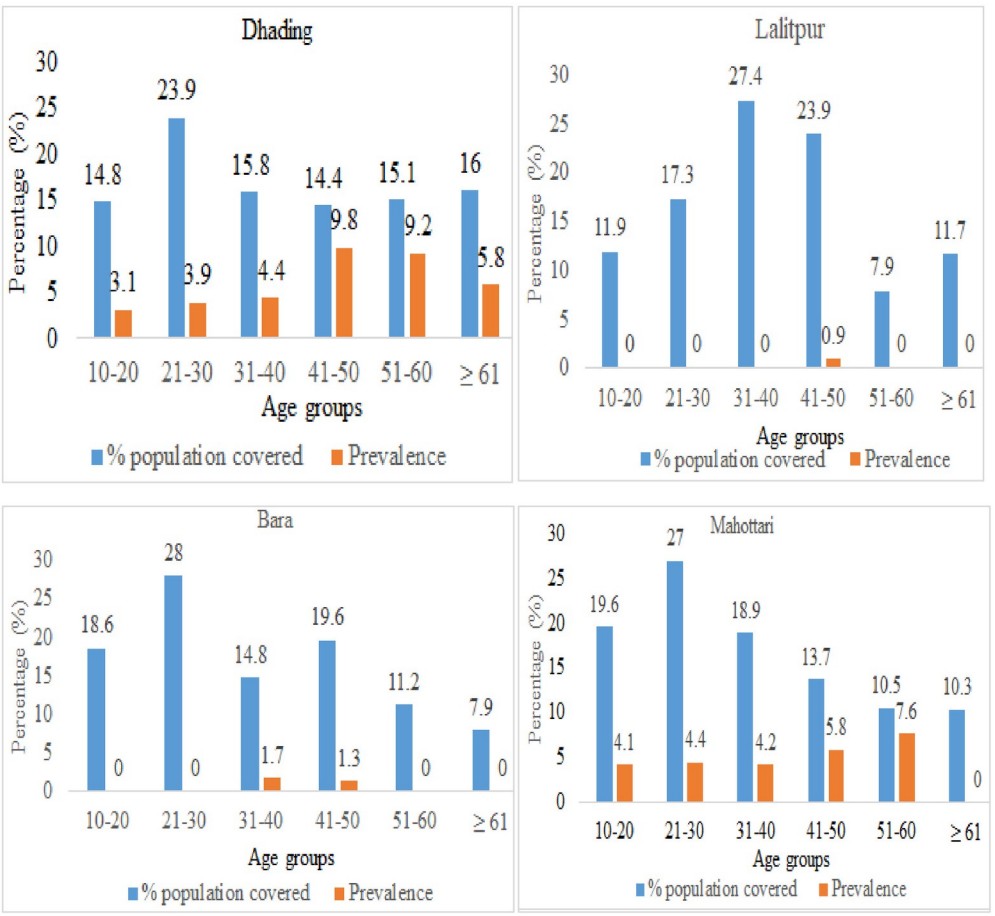

**Fig 3. Age-wise distribution of MF in hotspots of four districts of Central Nepal.**

participants were found more than 65% in the last round but in the hotspot of Mahottari district coverage was below 65% in previous MDA rounds. MF prevalence was found significantly high among individuals who have not participated in the both last and previous rounds of MDA except for drug uptake during the last round of MDA in Dhading district.

MDA compliance and the chronic clinical manifestation were not found to be remarkably associated. None of the individuals with hydrocele were found to be infected with MF but two individuals from the Dhading district were found to be infected with MF, who had never participated in the MDA program (Table 4).

## Discussions and conclusion

Under the global program to eliminate lymphatic filariasis, the Nepal government started the MDA program with the combination of DEC and albendazole in 2003. The program was launched in all sixty-three endemic districts, and TAS analysis was performed. Based on TAS indicating antigen prevalence < 2%, the MDA program has completely stopped in 51 districts including Lalitpur and Dhading districts of the hilly region and Bara and Mahottari districts of the Terai region. Earlier studies indicated that school-based TAS is not sufficient to stop MDA due to some limitations [9,10,20]. TAS only reflect the new infection in the community but community-based residual infection of MF plays a crucial role in future resurgence [21]. We analyzed the data of the baseline survey, Pre TAS, TAS I and TAS II in four districts. The

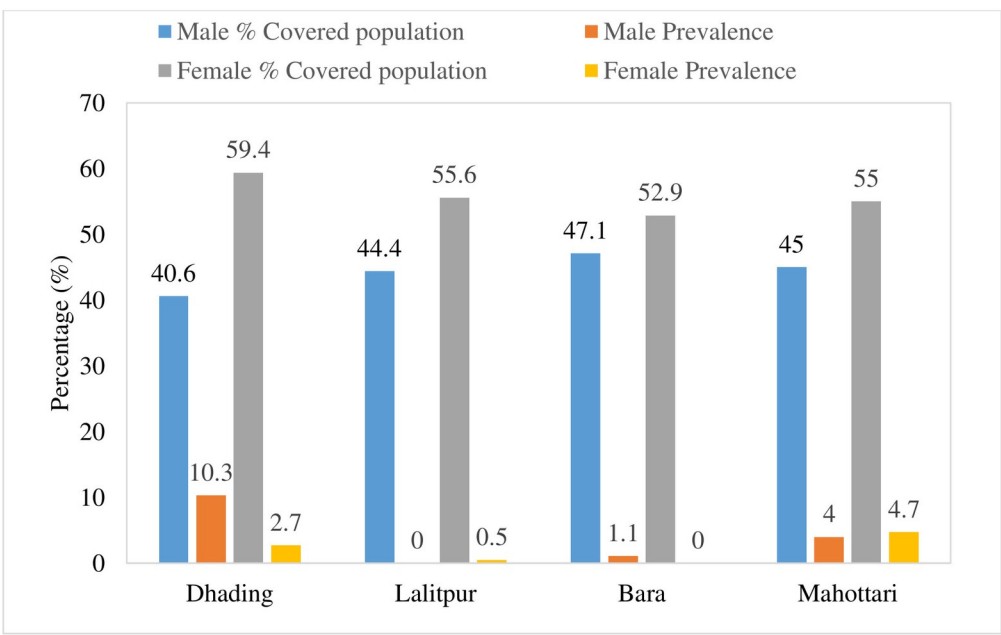

**Fig 4. Sex-wise distribution of MF in hotspots of four districts of Central Nepal.**

highest baseline prevalence was shown in Dhading district followed by Mahottari districts but in all TAS, CFA prevalence showed below critical level including other districts and MDA was stopped. Further analysis of data revealed interestingly CFA cases clustered in distinct localities which were considered hotspots. The clusters of antigen-positive children could be proxy indicators of residual microfilaremia carriers in community people of hotspots.

Community children who were born after the MDA program in study areas have reached around 10 years during the current study and they had never been participated in the program. Around 75% population out of 2285 eligible populations above 10 years were sampled for the MF screening in hotspots of four districts.

**Table 4. Site-based district-wise association of MDA compliance with microfilaria infection and chronic clinical manifestation of LF.**

| Districts | Never participated in MDA rounds | No. (%) total | No.(%) with MF | X² Value (p-value) | No. (%) with hydrocele* | Z- Value (p-value) | No. (%) with elephantiasis | X² Value (p-value) |
|---|---|---|---|---|---|---|---|---|
| Lalitpur[#] (n = 394) | Previous MDAs | 106 (26.9) | 1 (0.9) | - | 14 (29.2) | 3.05 (0.001) | 10 (10.0) | 1.018 (0.313) |
| | Last MDA | 99 (25.1) | 1 (1.01) | - | 9 (19.6) | 1.22 (0.213) | 6 (6.1) | 0.191 (0.662) |
| Dhading[β] (n = 431) | Previous MDAs | 110 (25.5) | 11(10.0) | 4.14 (0.042) | 14 (30.4) | 2.22 (0.024) | 11 (10.0) | 1.8 (0.18) |
| | Last MDA | 87 (20.2) | 8 (9.2) | 2.00 (0.157) | 14 (40.0) | 3.10 (0.001) | 7 (8.1) | 2.0 (0.157) |
| Bara[γ] (n = 393) | Previous MDAs | 83 (21.1) | 2 (2.4) | - | 4 (10.0) | 1.47 (0.068) | 2 (2.4) | 0.228 (0.633) |
| | Last MDA | 78 (19.8) | 2 (2.6) | - | 4 (10.3) | 1.49 (0.062) | 2 (2.6) | 0.327 (0.567) |
| Mahottari[α] (n = 504) | Previous MDAs | 189 (37.5) | 13 (6.9) | 4.2 (0.042) | 3 (3.6) | 1.34 (0.144) | 5 (2.7) | 8.2 (0.004) |
| | Last MDA | 174 (34.5) | 14 (8.1) | 7.8 (0.005) | 3 (3.8) | 1.39 (0.123) | 4 (2.3) | 4.5 (0.034) |

*Denominator included only males

[#] Dukuchhap and Bungmati of Lalitpur district

[β]Salyantar of Dhading district

[γ]Ammadar and Khairawa of Bara district

[α] Matihani of Mahottari district.

Treatment coverage and MDA rounds are crucial components of the LF elimination program. For the MDA program to be effective, high treatment coverage is crucial to meet the elimination target within the rational time frame [22]. A further experimental study suggested that endemic areas where the baseline prevalence of LF is high need high treatment coverage along with the increased number of MDA rounds [23]. However, it seems that in our study districts, only treatment coverage and TAS were considered. In the Dhading and Mahottari districts where baseline prevalence was comparatively high but MDA was stopped only after six rounds whereas in the Lalitpur and Bara districts MDA program was extended up to eight and eleven rounds respectively. We found high MF prevalence in those districts where baseline prevalence was also high and low MDA rounds were applied such as Dhading and Mahottari districts. Considering the persistence of cluster of LF positive cases as hotspots, night blood MF survey of ∼500 community individuals in those foci needs to be carried out for further identification of LF cases and foci treatment of all individuals in the hotspots is recommended. Hotspots of such districts might need bi-annual treatment along with improved treatment coverage and an increase in the number of MDA rounds [24].

Some studies on the impact of the MDA program on LF transmission and infection in areas where DEC and albendazole medicines were implemented such as in India, Papua New Guinea, Egypt, and American Samoa had documented good progress towards elimination [25–27]. In Nepal, most of the districts showed good progress in the MDA program as we observed in two districts Lalitpur and Bara. We observed LF positive cases in hotspots of two districts Dhading and Mahottari with high MF prevalence in age group 41–60. But sex-wise MF prevalence was not uniform in between the hilly district (Dhading) and Terai district (Mahottari). The high prevalence of MF in males of Dhading district could be due to the frequent national and international economic migration of males while in the Terai region migration could be comparatively less due to sufficient agricultural land and industrial development. Frequently migrating people often miss the drug uptake during the MDA program. However, some of the studies documented that optimal drug uptake during MDA is challenging in LF elimination in most of the endemic areas [28–30]. In the present study, community people's involvement in any previous round of the MDA program showed comparatively less than their involvement in the last round. The reason could be due to the recall bias as well as some of them had only participated in the last round after returning from economic migration. We found a significantly high MF prevalence in people who had not participated in previous and last MDA rounds.

The probability of LF infection in the hotspots and subsequent community seems to be related to the upper confidence level of MF prevalence at a 95% confidence interval (CI). We found an upper confidence level of LF prevalence above critical value in all hotspots of four districts indicating possible future resurgence. It has been recommended that the upper 95% CI of community CFA prevalence of 2% and School children antibody prevalence of 5% could be provisional critical cut-off values for the elimination of LF rather than the CFA critical cut–off value of TAS in school children [21]. The present study highlights the importance of community-based MF prevalence particularly among the 41–60 years age group less than 1% at 95% upper confidence interval need to be included while deciding provisional cut-off value.

The higher number of chronic clinical manifestations of LF such as hydrocele and elephantiasis cases in the community indicates the existence of LF infection for a long time. Evidence suggests that the cases are generally found in localized form. We were interested to know the MF infection among those populations sampled in the hotspots which could reflect the probable infection resurgence. Out of 68 hydrocele individuals in hotspots of four districts none of them were found infected with MF. Out of 70 elephantiasis individuals in the age group 41–60 years, two individuals tested positive for MF infection. Both of them were from Dhading

district and had not participated in the last MDA round which is a clear indication of LF resurgence in the hotspots of Dhading district. Similar results have also been reported in earlier studies indicating a low level of microfilaremia in individuals with elephantiasis and hydrocele [31,32].

Hence it is concluded that the number of MDA rounds depends on the baseline prevalence of LF infection in a particular community. Evaluation of an upper confidence interval of less than 1% at 95% CI is a crucial cut-off value at hotspots to stop MDA during the elimination program is important. During the TAS, the MF survey in the age group 41–60 years needs to be included before deciding to stop MDA.

## Supporting information

**S1 Fig. Patient with Lymphoedema.**
(TIF)

**S2 Fig. Sample collection by finger prick method for microfilaremia study in the field.**
(TIF)

**S3 Fig. Stained blood in stain rack.**
(TIF)

## Acknowledgments

We are thankful for the study area residents who were involved in this research. We also thank the Epidemiology and Disease Control Division and State Public Health Laboratory, Madhesh Pradesh Janakpurdham, Nepal. We express our deepest thanks to all Female Community Health Volunteers and all technical staff for helping in the collection of blood samples. We would like to show our gratitude to all elected heads of local governments of study areas of Central Nepal for their support in conducting the research. We also immensely thank Dr. Jagannath Adhikari, Birendra Multiple Campus, Chitwan, Pradip Rimal, EDCD, Kathmandu and Rambalak Rai, VBDRTC, Hetauda, Nepal for technical and logistic support.

## Author Contributions

**Conceptualization:** Pramod Kumar Mehta, Mahendra Maharjan.

**Data curation:** Pramod Kumar Mehta, Mahendra Maharjan.

**Formal analysis:** Pramod Kumar Mehta, Mahendra Maharjan.

**Funding acquisition:** Pramod Kumar Mehta.

**Investigation:** Pramod Kumar Mehta.

**Methodology:** Pramod Kumar Mehta, Mahendra Maharjan.

**Project administration:** Pramod Kumar Mehta, Mahendra Maharjan.

**Resources:** Pramod Kumar Mehta, Mahendra Maharjan.

**Software:** Pramod Kumar Mehta, Mahendra Maharjan.

**Supervision:** Mahendra Maharjan.

**Validation:** Pramod Kumar Mehta, Mahendra Maharjan.

**Visualization:** Pramod Kumar Mehta, Mahendra Maharjan.

**Writing – original draft:** Pramod Kumar Mehta.

**Writing – review & editing:** Pramod Kumar Mehta, Mahendra Maharjan.

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
