## [Decision Letter · Decision Letter 0]

10 Jul 2023

Dear Mr. Mehta,

Thank you very much for submitting your manuscript "Assessment of microfilaremia in ‘hotspots’ of four lymphatic filariasis endemic districts of Central Nepal" for consideration at PLOS Neglected Tropical Diseases. As with all papers reviewed by the journal, your manuscript was reviewed by members of the editorial board and by several independent reviewers. In light of the reviews (below this email), we would like to invite the resubmission of a significantly-revised version that takes into account the reviewers' comments. 

We cannot make any decision about publication until we have seen the revised manuscript and your response to the reviewers' comments. Your revised manuscript is also likely to be sent to reviewers for further evaluation.

Sincerely,

Jeremiah M. Ngondi, MB.ChB, MPhil, MFPH, Ph.D

Academic Editor

Francesca Tamarozzi

Section Editor

Reviewer's Responses to Questions

**Key Review Criteria Required for Acceptance?**

**Methods**

-Are the objectives of the study clearly articulated with a clear testable hypothesis stated?

-Is the study design appropriate to address the stated objectives?

-Is the population clearly described and appropriate for the hypothesis being tested?

-Is the sample size sufficient to ensure adequate power to address the hypothesis being tested?

-Were correct statistical analysis used to support conclusions?

-Are there concerns about ethical or regulatory requirements being met?

Reviewer #1: - The study objective is clearly articulated but the authors may have to consider consistency in the description of the study objective in various sections: Abstract (line 15,16), Author summary (49-51) and Introduction (90-92).

- The study rightly assesses post-treatment lymphatic filariasis (LF)/W. bancrofti infection by nocturnal microfilaremia(mf) (16,17, 125-127). 

- The study compares the mf prevalence with LF baseline, pre-TAS and TAS results to make a conclusion of increasing or decreasing infection. However, the study populations of above 9 years (115) is different from the comparison surveys: baseline survey used above 15 years (205-208), pe-TAS is usually uses above 5 years, TAS is usually in 6-7 years. Similarly, all the comparison surveys tested for filarial antigen (206, Table 4) compared to the mf test conducted in the study. The authors indicate that the selection of study age group was based on WHO recommendation (reference 18). However, a review of this reference did confirm this. WHO 2011 LF M&E guidelines age group recommendation for pre-TAS is > 5 years and TAS is 6-7 years. 

- The authors investigated suspected infection hotspots/persistent transmission sites. Hotspots was defined as foci with "a cluster of antigen-positive cases and mf carriers" after pre-TAS and TAS, but cluster of positive antigen cases or mf cases was not defined. A clear definition of number of antigen or mf positives that constitute a cluster and therefore hotspot to merit further investigation after a district has passed pre-TAS or TAS would 1) allow for reproducibility of the study 2) be helpful for identifying priority sites/clusters/communities for post-treatment or post-validation surveillance. 

- Study site: The authors stated random selection of survey participants from households, it would be important to readers how the households were selected, and what units the households were selected from e.g., community, sub-district, health area etc. This information is not provided. 

- The sample size is adequate as it is comparable to site sample size for pre-TAS

- Ethical considerations were adequately covered.

Reviewer #2: The study is a very important one for the global lymphatic filariasis (LF) community as there is little published research on how to detect and respond to areas of ongoing transmission after mass drug administration (MDA) ends. However, the article and the hypotheses statements do not articulate this framing well. For example, it is unclear throughout how the authors define hotspots and whether they are equivalent to districts or sites/villages. The methodology section of the article needs further details, including how sites were selected, how households within sites were selected, and the sampling framework and data source(s). If sites were selected based on pre-TAS or TAS data, it would be useful to include a brief explanation of those surveys in the paper's background section as well. The question of past participation in MDA is becoming more important and a global effort is ongoing to collect this data. As such, it would be helpful to include the exact questions that were asked about past participation in MDA. It would be useful to know what case definitions were used for hydrocele and elephantiasis and the qualifications of those who examined the participants. In terms of ethics, it would be helpful to know what information/treatment was given to participants with clinical conditions.

Reviewer #3: - This is acceptable. But the English needs editing to improve the content and description.

- Should a T-test be used instead of a chi square test?

**Results**

-Does the analysis presented match the analysis plan?

-Are the results clearly and completely presented?

-Are the figures (Tables, Images) of sufficient quality for clarity?

Reviewer #1: - Analysis matches the analysis plan.

- Results are clearly presented

- Figures and tables are of sufficient quality. Titles of tables 2,3 and 5 may be revised to include LF morbidity (lymphedema and hydrocele). Figure 2 may be clearer with a group bar charts for the 4 districts by year.

Reviewer #2: The data analysis would be more useful at a site, rather than district level, as disease prevalence and past participation in MDA can vary widely at village level. The analysis of 'trends' is inappropriate and should not be included, given that previous surveys all had different methodologies, diagnostic tests, age groups, etc. these prevalence levels cannot be compared. Instead the current cross-sectional prevalence can be compared to the microfilaremia cut off of 1%. In addition, given the issues with economic migrants in Nepal, it would be helpful to see an analysis of those 25% of individuals who did not enroll in the study, if that data was collected from the households. The methods mention that the study collected data on movement patterns but did not include this data in the results section - this is information lacking in the literature about Nepal and critical to the Nepal and India LF programs and should be included. 

Following are comments on the tables and figures:

- Figure 2 is not necessary. Would recommend combining the reported coverage as part of a revised Table 4 that includes background information on all districts. This could then be shifted to Table 1 as part of the background of these four districts, not to examine trends but to situate this study in the context of other LF information about the districts. This table could include population; baseline prevalence, diagnostic test and year; pre-TAS 'prevalence' diagnostic test and year; TAS diagnostic test, year, critical cut off and number of positives; and reported MDA coverage by year. 

- Current Table 1 would be more useful with rows for each site, and columns for 'eligible' and sampled populations, disaggregated by sex and age group. I would consider having a table with demographic characteristics and a separate table(s) with X2 and p-values. 

- Current Table 5 would be helpful to include results by site, as well as breakdown of those treated in last MDA and ever treated by age group and sex as well. Past participation results are usually analyzed by asking about participation in the most recent MDA and in all MDAs, including the most recent. If this was also how the authors analyzed it, their results of 75% in most recent and 72% ever participated do not make sense.

Reviewer #3: - When describing the mf prevalence, it will be useful to provide the percentage per district in the text, in addition to the Table that was provided.

- It would be useful to break the age groupings further to assess whether there were infections in children less than 9 years, and also to show the infection levels in different age categories. e.g. <10, 10-20, 21-30, 31-40, etc... The findings should also be discussed appropriately.

- Check Tables 3 and 4. The mf prevalence in Mahottari is indicated as 4.4 in one Table and 5.4 in the other Table. 

- Table 4. *, symbols indicate antigen prevalence above the critical value. there was no mention of assessing antigen prevalence in the methods or results. This is not clear and should be clarified, as it leaves me confused.

**Conclusions**

-Are the conclusions supported by the data presented?

-Are the limitations of analysis clearly described?

-Do the authors discuss how these data can be helpful to advance our understanding of the topic under study?

-Is public health relevance addressed?

Reviewer #1: - The mf prevalence recorded in two of the study sites surveyed in Dhading (5.8%) and Mahottari (5.4) would be concerning to the national program as they are above the expected threshold > 1%. This is an important finding. However, since the survey focused on suspected high prevalence/"hotspots" sites (Figure 1) in districts in post-treatment surveillance phase the authors may limit conclusions to the survey sites/communities rather than the districts (175,176, 231-233).

- The authors observe increasing trend of mf in two districts and otherwise in two other districts (181-184, Table 4). However, this is unsupported by the data presented 1) only one data mf prevalence data point is presented out of the 4 data points 2) data in Table 4 does show increasing mf or antigen prevalence trend for any of the four districts.

- Line 53-55: Authors may need to show how the study findings support recommendation for Xenomonitoring and antigen testing. 

- The limitation of the study and analysis were not presented.

Reviewer #2: As mentioned above, analysis of trends is not appropriate to do. Instead, the analysis, discussions and conclusion should focus on who is infected, how that links to past MDA participation, and recommendations on how the national program should respond. In addition, it would be useful to the global community to know the authors' recommendations on how to find 'hotspots', how to investigate them and how to respond in a programmatic context. No limitation section is included - it would be worthwhile to add this to the discussion.

Reviewer #3: No conclusions were provided in the discussion

**Editorial and Data Presentation Modifications?**

Reviewer #1: - Authors to look at consistency in use of LF infection, LF cases

- Effective MDA treatment coverage: ≥65% of total population. 

- Duration of last MDA and study may affect recall of question on participation in last MDA (187-189). Not clear time of study and last MDA. 

- Reconcile when LF program, MDA started, MDA stopped- 2003/2001, 2017, 2018

- Significance of mf prevalence by sex, age (≥ 41 years); lymphedema by age group: review interpretation of results (169,170) and Table 2 (p-values)

Reviewer #2: This article could benefit from an editor. Given that and the other major revisions stated above, it is not worthwhile to include all the minor modifications here that are needed.

Reviewer #3: - Significant language editing is required to make the paper acceptable for publication.

**Summary and General Comments**

Reviewer #1: The study is important for efforts to leave no foci of infection behind as countries make progress towards elimination of LF as a public health problem. The study is also important as a guide for country programs to target sites for post-treatment and post-validation surveillance. A clear definition of what constitutes a cluster of positive cases of concern to merit further investigation for persistent infection as in this study is fundamental to the manuscript. The authors also need to ensure that all conclusions are supported by the study. Some level of editing for consistency would enhance understanding of readers and flow.

Reviewer #2: The study represents an important entry into the global community about how to detect and respond to areas of potential ongoing transmission after MDA is stopped. However, more details are needed about methodology to understand how they detected these areas, and how they sampled the population. The analysis of the results by comparing with previous surveys is inappropriate. More analysis could be done about who was infected (looking at age, sex, occupation, movement patterns) and how past participation linked to infection status. It would be useful to include a discussion of how to respond to these areas and how the Nepal and other programs might translate this research into programmatic activities. With these major revisions, the study would be significant to the LF community as many programs move into post-MDA surveillance.

Reviewer #3: The authors assessed the mf prevalence in hotspot districts in Nepal, after the cessation of MDA. The findings of the study are interesting and the paper is of importance to the field and sustaining the LF elimination achievements. However, there are a number of issues that need to be addressed before it can be acceptable for publication.

- A strong language editing is recommended

- Kindly provide a more accurate description of the timelines in the abstract and main text. In the abstract, if MDA was started in 2001, that gives 15 rounds of MDA by 2016 and not 6 rounds of MDA. In the methods there is mention of 6 to 11 rounds of MDA from 2007 to 2017. These two different description of the MDA timelines are conflicting.

- Please present the mf data in the abstract.

- Kindly discuss the impact of recall bias on the assessment of MDA participation.

- Under the discussion, a concluding paragraph of the study will be useful.

- Given the results, kindly discuss the limitations of TAS in the decision to stop MDA and what can be done to improve the TAS processes.

PLOS authors have the option to publish the peer review history of their article (what does this mean?). If published, this will include your full peer review and any attached files.

Reviewer #1: No

Reviewer #2: No

Reviewer #3: No
---

## [Decision Letter · Decision Letter 1]

13 Dec 2023

Dear Mr. Mehta,

Thank you very much for submitting your manuscript "Assessment of microfilaremia in ‘hotspots’ of four lymphatic filariasis endemic districts of Nepal during post-MDA surveillance" for consideration at PLOS Neglected Tropical Diseases. As with all papers reviewed by the journal, your manuscript was reviewed by members of the editorial board and by several independent reviewers. The reviewers appreciated the attention to an important topic. Based on the reviews, we are likely to accept this manuscript for publication, providing that you modify the manuscript according to the review recommendations. 

I particular Reviewer #2 indicated that a number of aspects still need clarification, especially regarding the definition of hotspot and the sample size and sampling methodology.

These aspects need addressing.

Sincerely,

Jeremiah M. Ngondi, MB.ChB, MPhil, MFPH, Ph.D

Academic Editor

Francesca Tamarozzi

Section Editor

Reviewer's Responses to Questions

**Key Review Criteria Required for Acceptance?**

**Methods**

-Are the objectives of the study clearly articulated with a clear testable hypothesis stated?

-Is the study design appropriate to address the stated objectives?

-Is the population clearly described and appropriate for the hypothesis being tested?

-Is the sample size sufficient to ensure adequate power to address the hypothesis being tested?

-Were correct statistical analysis used to support conclusions?

-Are there concerns about ethical or regulatory requirements being met?

Reviewer #1: (No Response)

Reviewer #2: - The definition of hotspot is still slightly unclear. Lines 16-17 note that the TAS in sentinel sites was below threshold. Usually, sentinel sites are sampled in pre-transmission assessment surveys (pre-TAS), while TAS randomly samples ~30 clusters (schools or villages) in a district. The number of positive children in a TAS are compared to a critical cut off (roughly equal to 2% antigenemia) to see if MDA can be stopped. My guess is that TAS in these 4 districts passed, e.g. was less than the critical cut off, but certain clusters had multiple positive cases and these were classified as hot spots and erroneously called sentinel sites. However, this needs to be more explicitly defined in the paper and the abstract, specifically: was pre-TAS or TAS data used to determine hotspots? what constituted multiple positive cases, 2, 3, 4?

- Lines 118-120 The number of positives found in each cluster should be noted

- In the introduction, a short summary of TAS methodology would be appropriate to include around line 90-91. 

The specific sampling methods still requires more editing to be understandable. I would recommend discussing sample size and number of households at site level -currently it is discussed in totals, at district and site level and is difficult to follow. Table 2 helps capture some of this at district and total levels. However, if the sample size and the results are truly at site level, the other tables should also be constructed to show site level results.

- Line 135 Methods might flow better to start with required sample size and then discuss the number of households needed to be selected to meet this sample size.

- Line 139 I am not a sampling expert, but expecting 50% LF prevalence in a site (when known prevalence was nearer to 3-5%) seems to be inappropriate. 

Lines 185-189 Please include at which levels (site, district, total) these analyses were conducted.

Table 3 - results should be presented by site, not total

Reviewer #3: These are accpetable.

**Results**

-Does the analysis presented match the analysis plan?

-Are the results clearly and completely presented?

-Are the figures (Tables, Images) of sufficient quality for clarity?

Reviewer #1: (No Response)

Reviewer #2: The results are presented by total, by district and sometimes by hilly v Terai. As in the methods section, some clarity should be given in presenting at site level, by district and total. Perhaps a separate section and table could look at hilly v Terai if that is critical to the paper. (I'm not sure what the comparison of two districts within the Terai and two within the hilly region as in Table 3 tells us, other than districts/sites are very different.)

Table 3 - Consider bolding results which are significant so that the reader can easily see which ones were significant.

Lines 205-207 Phrasing such as 'much improvement' implies comparison with previous time points. Since that is not able to do be done due to previous different survey methodologies, rephrase to simply 'Mf prevalence is lower than the 1% cut off WHO recommends to stop MDA.'

Line 236 - 'MDA coverage' usually refers to community coverage. In this instance, I believe the 65% is referring to '65% of the study participants noted they participated in the last MDA.'

- Would recommend including a table that reports by site the percentage of participants who had never participated in any MDA and the percentage Mf positive of those who had never participated in any MDA. To answer this, I think the researchers will have to use only those participants who said no to participating in the last MDA round and those who also said no to participating in any other previous rounds.

Reviewer #3: These are acceptable

**Conclusions**

-Are the conclusions supported by the data presented?

-Are the limitations of analysis clearly described?

-Do the authors discuss how these data can be helpful to advance our understanding of the topic under study?

-Is public health relevance addressed?

Reviewer #1: (No Response)

Reviewer #2: - One complicating factor to this analysis is that Mahottari failed TAS3 in 2019, Bara failed TAS2 in 2017, and Laltipur was split into rural and urban areas for TAS. It's unclear when the data for this study were collected, but given that the article will be published in 2023/2024, they should address how their data links to the above survey results. E.g., line 116 notes that MDA has been stopped, which is not true at present.

- Line 270-271 is unclear to me - please revise.

- Lines 279-281 I believe some of the rounds in Lalitpur did not reach effective coverage (e.g. 65%) so did not count to the 5 rounds needed by WHO in order to move to pre-TAS and then TAS to stop MDA. In addition, please confirm the data on number of rounds before TAS1 - this does not match with other data. 

Lines 281-284 Note that WHO has not found evidence that biannual treatment is more effective than annual treatment (see Guidelines for LF treatment). Instead, I wonder if reported coverage was accurate or if there were groups of people not participating who continued to harbor microfilaremia. In addition, are you recommending that if a hotspot is found after TAS passes, that the entire district should restart MDA? Or are you recommending focal treatment be done in that hotspot only?

Line 321 - I would not link baseline prevalence in Nepal to number of rounds needed, given issues with baseline prevalence data collection in Nepal, as well as varying reported and surveyed coverage in individual MDA rounds. Your study wasn't designed to study that association.

Reviewer #3: These are acceptable.

**Editorial and Data Presentation Modifications?**

Reviewer #1: (No Response)

Reviewer #2: Line 49 - spell out Wuchereria

Line 53 - delete 'in sentinel sites'

Line 54 - not sure what is meant by 'residual' - presumably these were the first time the participants were tested for antigen. Delete.

Line 57 - No evidence this is resurgence of infection if we don't have comparable data. instead 'prevalence above threshold'

Line 83 - CFA prevalence during TAS hasn't been used in modeling to determine number of rounds of MDA needed

Lines 113-118 and Table 1 - TAS results should be presented as number positive vs a critical cut off and not as a percentage positive, as due to survey design creating a prevalence from the survey needs a special analysis

Table 1 - Please confirm - I believe baseline prevalence was Mf (and not antigen) in Bara, lalitpur, and Mahottari. 

Line 289 - The use of 'resurgence' is likely not appropriate since we don't have Mf data from a similar age group from those sites.

Reviewer #3: (No Response)

**Summary and General Comments**

Reviewer #1: Comments on the first draft have been adequately addressed. The readability of the manuscript has been improved. I

Reviewer #2: In general, the article includes very interesting data but more work is still needed to articulate the definition of hotspots, how the sampling was done, and present the results by site/district/total.

Recommending to do night blood sampling of adults in TAS (cluster based surveys of >1000 people) is likely a recommendation very difficult for most programs to implement. Are there recommendations that could address how to identify and follow up potential hotspots? Do you think it's feasible to do community night blood surveys of ~500 people in hotspots instead? And what would you recommend the program do for follow up, beyond just treating those who were found positive?

Reviewer #3: This is a much improved version of the manuscript. My comments have been addressed. Some minor grammatical errors and language editing is still required. However, I believe these will be covered during the proof stage before publication.

PLOS authors have the option to publish the peer review history of their article (what does this mean?). If published, this will include your full peer review and any attached files.

Reviewer #1: No

Reviewer #2: No

Reviewer #3: No

Figure Files:

Data Requirements:

Reproducibility:

References

---

## [Editor Report · Decision Letter 2]

22 Jan 2024

Dear Mr. Mehta,

We are pleased to inform you that your manuscript 'Assessment of microfilaremia in ‘hotspots’ of four lymphatic filariasis endemic districts of Nepal during post-MDA surveillance' has been provisionally accepted for publication in PLOS Neglected Tropical Diseases.

Best regards,

Jeremiah M. Ngondi, MB.ChB, MPhil, MFPH, Ph.D

Academic Editor

Francesca Tamarozzi

Section Editor

---

## [Editor Report · Acceptance letter]

29 Jan 2024

Dear Mr. Mehta,

We are delighted to inform you that your manuscript, "Assessment of microfilaremia in ‘hotspots’ of four lymphatic filariasis endemic districts of Nepal during post-MDA surveillance," has been formally accepted for publication in PLOS Neglected Tropical Diseases.

Best regards,

Shaden Kamhawi

co-Editor-in-Chief

Paul Brindley

co-Editor-in-Chief
